# Nonlinear dose-response association of moderate-to-vigorous physical activity with hyperuricemia in US adults: NHANES 2007–2018

Xia Zeng[1,2ஐ], Jitian Huang[1ஐ], Tianran Shen[2], Yingxia Xu[1], Xiaofang Yan[1], Qian Li[1], Yanmei Li[1], Xiaohui Xing[1], Qingsong Chen[3]*, Wenhan Yang[1,2]*

1 Department of Child and Adolescent Health, School of Public Health, Guangdong Pharmaceutical University, Guangzhou, Guangdong Province, China, 2 Department of Nutrition and Food Health, School of Public Health, Guangdong Pharmaceutical University, Guangzhou, Guangdong Province, China, 3 Department of Occupational Health, School of Public Health, Guangdong Pharmaceutical University, Guangzhou, Guangdong Province, China

ஐ These authors contributed equally to this work.
* wenhan-yang@gdpu.edu.cn (WY); qingsongchen@aliyun.com (QC)

**Data Availability Statement:** The original contributions presented in the study are publicly available. The datasets analyzed during current study are available at NHANES online website: https://www.cdc.gov/nchs/nhanes/index.htm.

## Abstract

### Background

The relationship between physical activity and hyperuricemia (HUA) remains inconsistent, and the dose-response association between moderate-to- vigorous physical activity (MVPA) level and HUA still unclear. In this study, we aimed to investigate the dose-response association of MVPA with HUA, and to explore an appropriate range of MVPA level for preventing HUA.

### Methods

Data from the US National Health and Nutrition Examination Survey (NHANES) 2007–2018 were used, including 28740 non-gout adult Americans. MVPA level was self-reported using the Global Physical Activity Questionnaire and serum uric acid was measured using timed endpoint method. The dose-response relationship between MVPA level and HUA was modeled with restricted cubic spline analysis. Logistic regression analysis were applied to estimate odd ratios (ORs) and 95% confidence intervals (CIs) of the relationships between MVPA level and HUA.

### Results

A total of 28740 adults were included in the study (weighted mean age, 47.3 years; 46.5% men), with a prevalence rate of HUA was 17.6%. The restricted cubic spline functions depicted a general U-shaped relationship between MVPA level and HUA. The MVPA level of 933 and 3423 metabolic equivalent (MET) -min/wk were the cut-off discriminating for the risk of HUA. Participants with MVPA levels in the range of 933–3423 MET-min/wk had lower risk of HUA and they had the lowest risk when MVPA levels at around 1556 MET-min/wk.

**Funding:** For this study, Wenhan Yang was supported by the National Natural Science Foundation of China (Grant No. 81973063). The funder had no role in study design, data collection and analysis, decision to publish, or preparation of the manuscript.

**Competing interests:** The authors have declared that no competing interests exist.

Compared with the moderate-activity group (600–2999 Met-min/wk), the low-activity group (< 600 Met-min/wk) had a greater risk of HUA (OR, 1.13 [95%CI, 1.02–1.26]) after fully adjusting for potential confounders.

## Conclusions

Compared with the moderate MVPA level, the low MVPA level was associated with the higher risk of HUA. And there may be a U-shaped dose-response relationship between MVPA level and HUA. When MVPA level was approximately 933–3423 MET-min/wk, the risk of HUA may at a lower level and the risk reached the lowest when MVPA level at around 1556 MET-min/wk.

## Introduction

Hyperuricemia (HUA) is a metabolic disease. Purine metabolism disorder in the body decreased excretion or increased production of serum uric acid (SUA), which developed into HUA [1]. The prevalence and incidence of HUA in the world population have been continuously rising over the past four decades [2, 3], which has been considered as an emerging public health concern over the world. The prevalence of HUA was increased from 18.2% in 1988–1994 to 20.1% in 2015–2016 among US adults based on the data of the National Health and Nutrition Examination Survey (NHANES) in the US [4]. Similarly, a study revealed a 4.3% increase in the prevalence rate of HUA within 4 years (2005–2009) in Italy [5]. A systematic review and meta-analysis pooled the results of the studies at different time points (2000–2014) and showed that 13.3% of the Chinese population had HUA, with a tendency to become younger [6]. HUA has been confirmed as an important risk factor for gout. Furthermore, HUA is also one of the independent risk factors for several chronic diseases [7]. Studies showed the incidence of coronary heart disease, hypertension and diabetes respectively increased by 12% [8], 13% [9] and 17% [10] while the SUA level increases by 60mmol/L.

An effective measure to control HUA focuses on lifestyle behavioral adjustment including improving physical activity(PA) [11]. A study have reported that PA can reduce mortality risks from high SUA [12]. As PA is receptive and with much potential transient and long-lasting health benefits, compared to the medication [12]. A Korean cohort study indicated the individuals that more frequently participated in health enhanced physical activity (HEPA) had a lower HUA odds ratio than those with lower PA participation rate (OR = 0.90, 95% Ci = 0.86–0.93) [13]. Yuan S, et al. reported that the SUA of a 45-day aerobic PA program (1600-meter jogging) decreased by 10.5% ($p < 0.05$) [14]. However, previous research conclusion on the relationship between PA and HUA still remains inconsistent. A few epidemiological studies and small clinical trials reported that PA had no positive influence on the level of SUA [15–17]. What's more, other studies also showed that high PA increases the risk of high SUA [18, 19]. We speculated that these above inconsistent results may be caused by nonlinear association between PA and SUA. As far as we know, few studies had explored the dose-response relationship between PA and HUA at present [13] and there is no specific study like our research to investigate their dose-response association. It is meaningful and necessary to explore appropriate recommended PA level to prevent HUA.

Therefore, we used data from the NHANES of the US to investigate the dose-response association of moderate-to-vigorous physical activity (MVPA) with HUA, and to explore an appropriate range of MVPA level for SUA level in this study. We hypothesized that the does-

response association between MVPA level and HUA was nonlinear, and there were significant differences in the risk of HUA among groups with different levels of MVPA.

## Methods

### Study population and design

National Center for Health Statistics (NCHS) at the Centers for Disease Control and Prevention (CDC) organized a large project named National Health and Nutrition Examination Survey every 2 years [20]. It is a nationally representative survey of civilian, non-institutional United States population, which in order to evaluate the health and nutritional status of the U.S. population. NHANES used a complex, multistage, probability sampling design to ensure population representativeness. The survey was made up of an in-home interview followed by a standardized health examination in mobile medical examination centre with specialized equipment, including a physical examination and laboratory inspection by trained medical personnel [21, 22]. The NCHS Ethics Review Board has reviewed and approved the survey protocol (Protocol #2005–06; Protocol #2011–17; Protocol #2018–01). All respondents agreed the survey and their written informed consent was obtained. A detailed description of the surveys has been published elsewhere [23]. The Guangdong Pharmaceutical University Academic Review Board determined the present study was exempt from approval because of the use of deidentified data.

In the present study, we used the data from the NHANES 2007–2018, six cross-sectional survey cycles (2007–2008, 2009–2010, 2011–2012, 2013–2014, 2015–2016, and 2017–20018, respectively), because these cycles include data on the both the exposures and outcome of interest. We included 34770 individuals aged 20 years and older. Of these, 1475 participants were not included because they had gout. Besides, 1167 participants were excluded because they lacked complete self-reported data on MVPA levels or had abnormal data. In addition, 3388 participants were still excluded because they lacked data on SUA. Eventually, 28740 Americans were included in this study (**Fig 1**). Data on SUA levels were available from laboratory examination of serum specimens; MVPA levels were calculated based on the participants' self-reported information about Global Physical Activity Questionnaire(GPAQ) [24]. This analysis involving deidentified data with no direct participant contact was not considered to be human subjects research and was not subject to institutional review board review, based on National Institutes of Health policy.

### Measurements and variable definitions of moderate-to-vigorous physical activity

During the in-home interviews, trained medical personnel utilize some questions related to activity for work, during transport and leisure time, which were based upon the Global Physical Activity Questionnaire(GPAQ) to collected information on participants' MVPA level [25]. Participants self-reported the number of days that they took part in miscellaneous types of PA in a representative week. The activity classifications were consisted of 5 types of activities, which were vigorous work activity, vigorous leisure activity, moderate work activity, moderate leisure activity, and activity for transportation. Participants would be required to report the amount of time (in minutes) they spent doing a specific type of activity on a representative day if they reported to investigators that they doing any day of that type of PA. Then multiplied the number of days reported by the specific amount of daily minutes spent on that PA to calculate the total number of minutes per week for each type of PA. The metabolic equivalent (MET)-min/wk was calculated by multiplying total minutes per week of each activity by standard

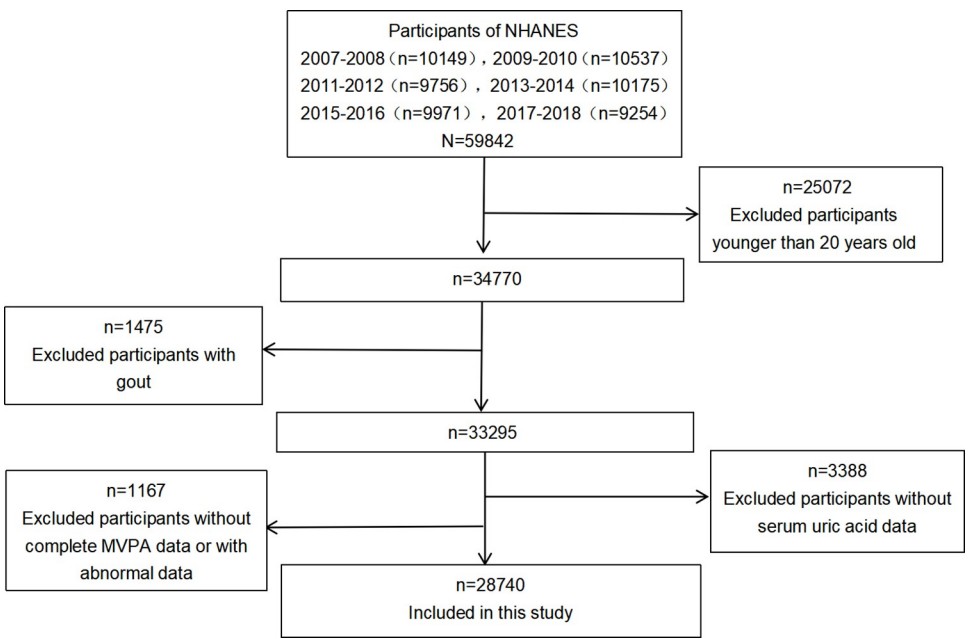

**Fig 1. Flow chart of the screening process for the selection of eligible participants.**

MET values recommended by NHANES [26, 27]. Eventually, summing MET-min/wk from all PA classification to get the total MET-min/wk for all activity [24]. Total activity in MET-min/week was classified into three groups of low (< 600 MET-min/wk), moderate (600–2999 MET-min/wk), and high (≥ 3000 MET-min/wk) activity [28].

## Measurements and variable definitions of hyperuricemia

The laboratory procedures and methods were described in details on the NHANES website (https://wwwn.cdc.gov/Nchs/Nhanes/2007-2008/BIOPRO_E.htm). Serum uric acid level (SUA)for 2007 was measured by Beckman Synchron LX20 using timed endpoint method, while for 2008–2014, this method is still used even if the equipment is replaced by DxC800 [29]. SUA from 2015 to 2018 was measured by DxC800 and DxC660i using time endpoint method. HUA was defined as SUA levels >7mg/dL in males and >6mg/dL in females in two fasting tests on different days under the condition of normal purine diet [7].

## Covariates

Socio-demographic variables [age, gender, race/ethnicity, education, and poverty/income ratio (PIR)] were considered as potential confounders. Ethnicity was categorized as Mexican American, other Hispanics, non-Hispanic white, non-Hispanic black, and other races (including multiracial) [30]. Education level was divided into below high school (including grade 12 without a diploma), high school, and higher than high school (e.g. university or higher) [31]. As a measure of socioeconomic status, PIR was categorized as different categories. The higher the PIR was, the better the household income situation [32]. Self-reported smoking (current smoker versus current non-smoker) and alcohol intake (no drinks, ≤1 drink per week, and >1 drink per week) were used to reflect other health behaviors [33]. Regarding health-related variables, body mass index (BMI) and gout were taken into account. Height and weight were

measured at mobile exam centers for the determination of BMI (normal: below $<25$ kg/m$^2$; overweight: between 25 and $<30$ kg/m$^2$; obesity: $\geq 30$ kg/m$^2$) [34].

## Statistical analysis

NHANES used a complex, multistage probability sampling design to represent the civilian non-institutionalized U.S. population. Therefore, all analyses were adjusted for survey design and weighting variables following the NHANES analytical guidelines [35].

In baseline characteristics of the samples, the Kolmogorov-Smirnov normality test was used to determine whether the continuous variables conform to the normal distribution and the mean±standard deviation was used to describe normally distributed variables. The chi-square test was adopted to compare the percentages of categorical variables among different MVPA groups.

The restricted cubic spline model was used to flexibly model the underlying dose-response association between continuous MVPA level and HUA, with 4 knots at the 5th, 35th, 65th and 95th percentiles of the MET-min/wk distribution. In the spline model, we adjusted for sex, age and race/ethnicity, education, PIR, smoking status, alcohol status and BMI.

Binary logistic regression model was used to estimate odds ratios (ORs) and 95% confidence intervals (CIs) between different MVPA level groups and HUA. The moderate group (600-2999MET-min/wk) was used as the reference in the models. Model 1 adjusted for age, sex and race/ethnicity. Model 2 added PIR and education level. Model 3 increased smoking and alcohol consumption and BMI.

Furthermore, we performed stratified analyses and interaction analyses to examine whether the association differed by sex, age, race/ethnicity and BMI while adjusting for the other variables and whether there was an interaction (S1 Table).

Results were shown as odds ratios (ORs) with 95% confidence intervals (CIs). A 2-sided P value less than 0.05 was considered statistically significant. Data files were downloaded from the NHANES website in October of 2021 and were processed and merged using SAS Version 9.4 (SAS Inst., Cary, NC, USA). The statistical analyses were conducted in the SAS and R Version 4.1.3.

## Results

Of 34770 adults aged at least 20 years at the time of the survey, 28740 did not suffer from gout and had valid self-reported MVPA data and SUA data, so that they were included in the analysis. These respondents were comprised 15345 women (weighted proportion, 53.5%) and 13395 men (weighted, 46.5%), with a weighted mean age of 47.3 years. The participants were categorized into five racial groups: Mexican American (n = 4468, weighted, 8.7%), Other Hispanic (n = 3098, weighted, 6.0%), Non-Hispanic White (n = 11754, weighted, 66.6%), Non-Hispanic Black (n = 5830, weighted, 10.7%), and the other race (n = 3590, weighted, 8.2%). The weighted mean SUA level was 5.36 mg/dL and the weighted mean BMI was 29.0 kg/m$^2$. Of the 28740 participants, 11839 (weighted, 36.2%) participants reported their MVPA levels were$<600$ MET-min per week, accounting for the largest proportion, followed by 8740 (weighted, 33.1%) participants reported their MVPA levels were between 600 and 2999 MET-min per week. Besides, 8161 (weighted, 30.7%) participants reported their MVPA levels were$>3000$ MET-min per week. Characteristics of the study population among different MVPA levels were summarized in **Table 1**.

There were significant differences among the three groups with respect to sex, age, race, ratio of family income to poverty, education, smoking and alcohol status and BMI ($p < 0.05$). Besides, we also found that compared with those who did less MVPA per week, participants

**Table 1. Descriptive characteristics of participants for total sample and by MVPA level groups among US adults[a].**

| Characteristics[b] | MVPA group, MET-min/wk, No. (%) | | | Total (N = 28740) | p value[c] |
|---|---|---|---|---|---|
| | Low: < 600 (n = 11839) | Moderate:600–2999 (n = 8740) | High: ≥ 3000 (n = 8161) | | |
| Age, mean (95%CI), y | 51.8 (51.3–52.3) | 46.8 (46.1–47.4) | 42.4 (41.8–43.0) | 47.3 (46.8–47.7) | <0.05 |
| BMI, mean (95%CI) | 30.0 (29.8–30.2) | 28.4 (28.2–28.6) | 28.4 (28.2–28.6) | 29.0 (28.8–29.2) | <0.05 |
| SUA, mean (95%CI) | 5.4 (5.3–5.4) | 5.3 (5.2–5.3) | 5.5 (5.4–5.5) | 5.4 (5.3–5.4) | <0.05 |
| **Sex, N(%)** | | | | | |
| Men | 4524 (36.9) | 3991 (44.6) | 4880 (59.8) | 13395 (46.5) | <0.05 |
| Women | 7315 (63.1) | 4749 (55.4) | 3281 (40.2) | 15345 (53.5) | |
| **Age, N(%)** | | | | | |
| 20–29 | 1250 (12.1) | 1513 (18.5) | 2050 (27.0) | 4813 (18.8) | <0.05 |
| 30–39 | 1584 (14.7) | 1612 (19.4) | 1720 (20.6) | 4916 (18.1) | |
| 40–49 | 1920 (18.5) | 1500 (18.6) | 1499 (19.5) | 4919 (18.8) | |
| 50–59 | 1962 (19.4) | 1452 (19.7) | 1255 (16.7) | 4669 (18.7) | |
| ≥60 | 5123 (35.2) | 2663 (23.8) | 1637 (16.3) | 9423 (25.6) | |
| **Ratio of family income to poverty, N(%)** | | | | | |
| ≤ 1.30 | 3736 (22.5) | 2124 (15.8) | 2371 (20.3) | 8231 (19.6) | <0.05 |
| 1.31–3.50 | 4061 (33.9) | 2765 (29.5) | 2876 (33.9) | 9702 (32.4) | |
| >3.50 | 2699 (34.4) | 3023 (47.4) | 2171 (38.1) | 7893 (39.8) | |
| Missing | 1343 (9.3) | 828 (7.3) | 743 (7.7) | 2914 (8.1) | - |
| **Race/ethnicity[d], N (%)** | | | | | |
| Mexican American | 1948 (9.4) | 1163 (6.8) | 1357 (10.0) | 4468 (8.7) | <0.05 |
| Other Hispanic | 1364 (6.5) | 846 (5.2) | 888 (6.2) | 3098 (6.0) | |
| Non-Hispanic White | 4555 (63.2) | 3768 (70.2) | 3431 (66.5) | 11754 (66.6) | |
| Non-Hispanic Black | 2529 (12.0) | 1629 (8.9) | 1672 (10.7) | 5830 (10.7) | |
| Other Race—Including Multi-Racia种族 | 1443 (8.9) | 1334 (8.9) | 813 (6.6) | 3590 (8.2) | |
| **Education level, N(%)** | | | | | |
| Less than high school | 3600 (20.6) | 1622 (10.9) | 1795 (14.9) | 7017 (15.6) | <0.05 |
| High school | 2768 (24.1) | 1674 (18.7) | 1952 (24.3) | 6394 (22.4) | |
| Higher than high | 5450 (55.2) | 5434 (70.4) | 4410 (60.7) | 15294 (61.9) | |
| Missing | 21 (0.1) | 10 (0.1) | 4 (0.1) | 35 (0.1) | - |
| **Smoking status, N(%)** | | | | | |
| Non current smoker | 2832 (24.6) | 2043 (24.3) | 1755 (22.6) | 6630 (23.9) | <0.05 |
| Current smoker | 2284 (19.2) | 1422 (15.1) | 2011 (24.0) | 5717 (19.3) | |
| Missing | 6723 (56.2) | 5275 (60.6) | 4395 (53.4) | 16393 (56.8) | - |
| **Alcohol status, N (%)** | | | | | |
| No drinks | 2457 (17.9) | 1248 (11.6) | 1069 (11.5) | 4774 (13.9) | <0.05 |
| ≤ 1 /week | 4593 (42.7) | 3777 (44.3) | 3782 (46.5) | 12152 (44.4) | |
| >1 /week | 1687 (17.7) | 1919 (27.5) | 1910 (28.0) | 5516 (24.1) | |
| Missing | 3102 (21.7) | 1796 (16.7) | 1400 (14.0) | 6298 (17.7) | - |
| **BMI, kg/m², N(%)** | | | | | |
| <18.5 | 208 (1.7) | 139 (1.5) | 108 (1.5) | 455 (1.6) | <0.05 |
| 18.5–24.9 | 2815 (23.2) | 2602 (30.5) | 2385 (31.1) | 7802 (28.1) | |
| 25.0–29.9 | 3742 (31.2) | 2946 (34.5) | 2706 (33.1) | 9394 (32.9) | |
| ≥ 30.0 | 4843 (42.2) | 2990 (33.0) | 2907 (33.8) | 10740 (36.6) | |

*(Continued)*

**Table 1.** (Continued)

| Characteristics[b] | MVPA group, MET-min/wk, No. (%) | | | Total (N = 28740) | p value[c] |
|---|---|---|---|---|---|
| | Low: < 600 (n = 11839) | Moderate:600–2999 (n = 8740) | High: ≥ 3000 (n = 8161) | | |
| Missing | 231 (1.6) | 63 (0.6) | 55 (0.6) | 349 (1.0) | - |

Abbreviations: BMI: body mass index (calculated as weight in kilograms divided by height in meters squared); MET: metabolic equivalent; MVPA, moderate-to-vigorous physical activity per week (minutes); SUA: serum uric acid.

[a] Data source: NHANES, National Health and Nutrition Examination Survey, 2007–2018.

[b] All means and SDs for continuous variables and percentages and SDs for categorical variables were weighted, with the exception of the number of participants. Because all numbers were rounded, percentages may not total 100%.

[c] p values were computed separately for each covariate and indicate statistically significant differences between step groups if $p < 0.05$.

[d] Race/ethnicity was determined using preferred terminology from the National Center for Health Statistics as non-Hispanic white, non-Hispanic black, and Mexican American. Mexican American individuals were oversampled rather than broader groups of individuals from Latin America. Other includes Asian, other Hispanic, Alaskan native, and multiracial individuals.

with higher weekly levels of MVPA were more likely to be male and significantly younger, and their BMI was also lower.

Restricted cubic spline model was used to evaluate the dose-response relationship between MVPA level and HUA. In full adjusted model (Fig 2), the MVPA level was associated with HUA. The dose-response association between them was nonlinear (p for nonlinear < 0.05) and presented a general U-shaped. With the increased of MVPA level, the risk of HUA gradually decreased and reached the lowest value when MVPA level was about 1556 MET-min/wk. With the continuous increased of MVPA level, the risk increased slowly and gradually leveled off. Two cut-off discriminating point of MVPA level were 933 and 3423 MET-min/wk.

In the primary results, compared with participants in the moderate MVPA level group, there was a significantly positive association of HUA for participants in low MVPA level group after adjustment for all covariates (OR, 1.13 [95% CI, 1.02–1.26]), but a similar relationship did not exist in the high PA group ($p > 0.05$) (Table 2).

Stratified analysis indicated that the association of low MVPA level with higher risk of HUA was only in Non-Hispanic White (OR, 1.22 [95% CI, 1.06–1.41]) and participants with obesity (OR, 1.25 [95% CI, 1.08–1.43]). But in terms of interaction analysis, sex and age made sense (S1 Table).

## Discussion

In this epidemiological survey conducted among American adults, we found a general U-shaped dose-response relationship between MVPA level and HUA. The MVPA level of 933 and 3423 MET-min/wk were the cut-off discriminating for the risk of HUA and the lower and higher MVPA level apparently increasing the risk. Participants whose MVPA level was in the range of 933–3423 MET-min/wk with lower risk of HUA and the risk reached the lowest when MVPA level at around 1556 MET-min/wk. In addition, we also found an increased risk of HUA (OR, 1.13 [95%CI, 1.02–1.26]) in the low-activity group (< 600 Met-min/wk) compared with the moderate-activity group (600–2999 Met-min/wk) after fully adjusting for potential confounders.

This study had shown that there were 11839 participants, or 41.2% of the surveyed population, didn't reached the physical activity standard which recommended by WHO [36] while an another available data suggest that 31.1% of the world's adult population is not meeting the minimum recommendations for physical activity [37]. So compared with the world average,

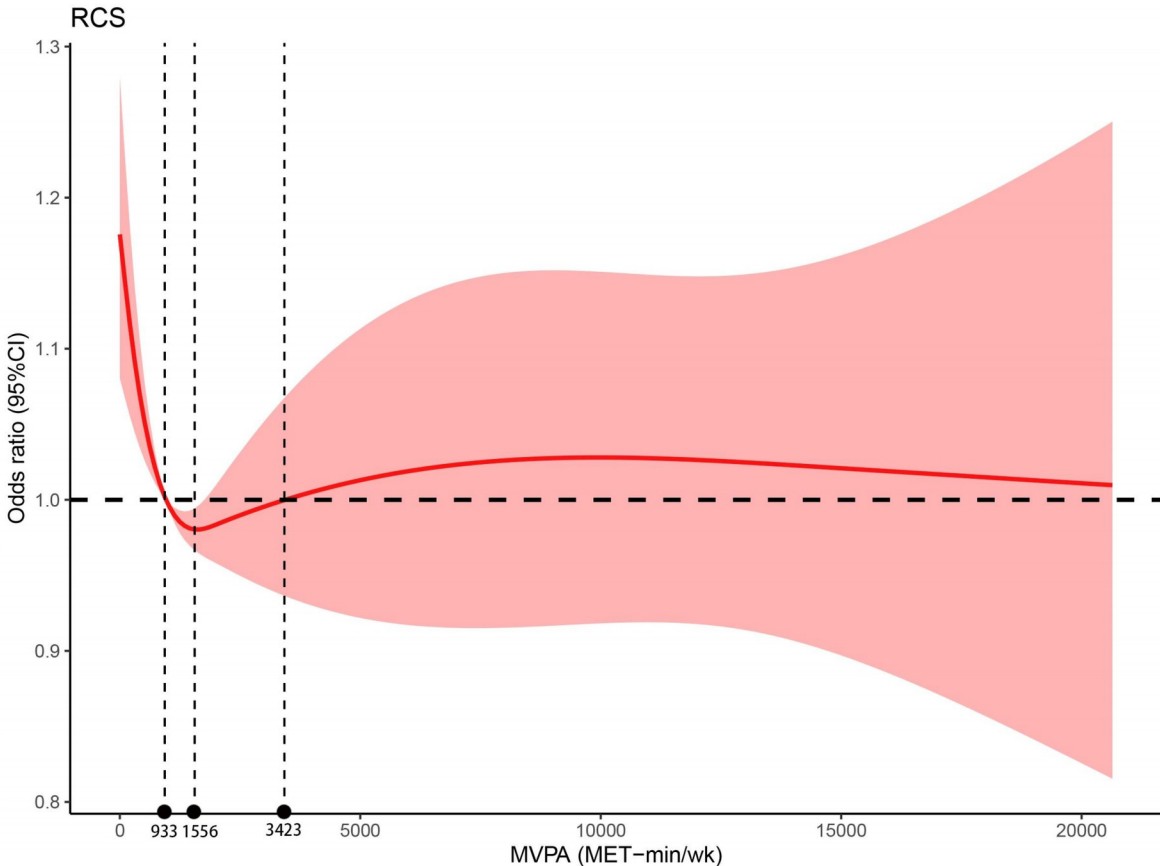

**Fig 2. Dose-response association of MVPA levels with HUA.** Abbreviations: MVPA, moderate-to-vigorous physical activity per week (minutes). Restricted cubic splines of odd ratios of MVPA level with HUA. The model was adjusted for age, race, sex, education, poverty income ratio, smoking status, alcohol status and body mass index. Restricted cubic splines were fit with 4 knots placed at the 5th, 35th, 65th and 95th percentiles of MVPA. Shading indicates 95% CI. The spline was nonlinear ($p$ for nonlinear < 0.001) and presented a general U-shaped. Two cut off value point was approximately 933 and 3423 MET-min/week and the level of MVPA corresponding to the lowest point was 1556 MET-min/week.

the study group had a higher rate of underachievement in physical activity. This was consistent with a number of studies suggesting that the standard-reaching rate of physical activity has become increasingly pessimistic in recent years as people's lifestyles have changed. Besides, the

**Table 2. Association of MVPA Level with HUA.**

| MVPA group, MET-min/wk | Case/Participants, No. | Odds ratio (95% CI) | | | |
| --- | --- | --- | --- | --- | --- |
| | | Crude Model | Model 1[a] | Model 2[b] | Model 3[c] |
| **Moderate: 600–2999** | 1399/8740 | 1 [Reference] | 1 [Reference] | 1 [Reference] | 1 [Reference] |
| **Low:** <600 | 2339/11839 | **1.31 (1.19–1.44)**[*] | **1.27 (1.15–1.39)**[*] | **1.24 (1.13–1.36)**[*] | **1.13 (1.02–1.26)**[*] |
| **High:** ≥ 3000 | 1307/8161 | 1.05 (0.95–1.15) | 1.05 (0.96–1.16) | 1.04 (0.94–1.14) | 1.06 (0.96–1.18) |

Abbreviations: MVPA, moderate-to-vigorous physical activity per week (minutes)

[a] Adjusted for age, sex, and race.

[b] Adjusted for variables included in model 1 + ratio of family income to poverty, personal highest education level.

[c] Adjusted for variables included in model 2 + alcohol, smoking, body mass index, hypertension

[*]: $p < 0.05$

number of participants with HUA in this survey was 5045, with a prevalence rate of 17.6%. This was slightly less than 20.1% in 2015–2016 among US adults based on the data of the NHANES in the US [4]. We speculated that this may be due to the exclusion of patients with gout in the study.

Our findings were in line with previous studies. Several studies had shown that low PA was associated with high SUA. A cohort study in Taiwan [12] found that low PA was associated with high SUA level, and adequate PA could potentially overcome high SUA level and more risks, which was similar to the results of this study. In addition, another cohort stud in Italy [38] also found that low PA increases the risk for higher SUA. Although the underlying mechanisms of PA and SUA remain unclear, several possible biological pathways have been proposed. First, PA can increase insulin sensitivity, which may be mediated the relationship between PA and serum uric acid level [39]. Secondly, long-term aerobic exercise can significantly increase serum phosphorus, and accelerate the conversion rate of ATP, reduce serum ATP and its metabolites (manifested as decreased ATP, adenosine, hypoxanthine), and thus reduce serum uric acid [14]. In addition, a study [40] reported that individuals who with chronic exercise, their plasma volume increases as much as 10% with an accompanying elevation of glomerular filtration rate and expansion of extracellular fluid volume. And an enhanced extracellular volume and subsequent improvement of renal plasma flow would enhance delivery of urate for secretion and subsequent excretion. These possible mechanisms explained to some extent that a certain level of PA can reduce SUA level. As for the amount of PA level, WHO previously recommended that adults should do an equivalent combination of moderate-to-vigorous physical activity achieving at least 600 MET-minutes [36] throughout a week, including activity for work, during transport and leisure time. However, there was no PA level recommendation has been made to prevent HUA. To our knowledge, this was the first time to explore the appropriate range and optimal MVPA level for preventing HUA.

The dose-response relationship between MVPA level and HUA obtained in this study is non-linear and may present U-shaped. With the increased of MVPA level, the risk of HUA decreased first and then gradually increased. Xiong Z, et al. [18] had reported that although low PA was associated with high SUA, continuous high intensity PA also improves SUA level and increases the risk of HUA. A study also reported that the excessive exercise may cause serum uric acid elevation [41]. The possible mechanisms that explained this phenomenon was that rigorous exercise consumes adenosine triphosphate at a greater rate than can be produced and resupplied to exercising muscles, leading to adenine nucleotide degradation, followed by increases in the plasma concentrations of hypoxanthine, xanthine, and urate. In addition, such physical activity also increases the concentrations of lactic acid in blood and noradrenaline in plasma and also decreases creatinine clearance, leading to a decrease in the urinary excretion of uric acid, xanthine, and oxypurinol [16]. However, previous studies[12, 13] did not discuss or specify a upper threshold for PA. Interestingly, our study had explored the upper limit of MVPA level for preventing HUA, which is the first time to explore the high cutoff value in the research on the relationship between MVPA level and HUA.

In this study, we used a large nationally representative sample of data from NHANES, which allowed us to generalize our findings to a wider population. However, there are some important limitations in this study. First, due to the cross-sectional study design, the ability to infer a causal relationship between MVPA level and HUA was limited. Secondly, NHANES collected self-reported MVPA levels of respondents through questionnaires, which may be unable to avoid recall bias and reporting bias, because PA is a socially desirable behavior and people may tend to over report it [12]. In addition, although we adjusted for many potential confounding factors, we could not rule out the possibility of residual confounding due to

unmeasured or unidentified factors, such as medication taking. Finally, Post-Hoc tests was needed in statistical analysis to understand which specific MVPA level group is different.

## Conclusions

In conclusion, this study based on a nationally representative population of American adults found that compared with the moderate MVPA level, the low MVPA level was associated with the higher risk of HUA. There was a general U-shaped dose-response relationship between MVPA level and HUA. When MVPA level was approximately 933–3423 MET-min/wk, the risk of HUA may at a lower level and the risk reached the lowest when MVPA level at around 1556 MET-min/wk. It is important for adults to have a certain level of MVPA to prevent HUA. Furthermore, the association we investigated in this study is plausible, but further large scale prospective studies are required to confirm the causal relationship between MVPA level and HUA.

## Supporting information

**S1 Table. Stratified and interaction analyses for the association of MVPA levels with HUA[a].** Abbreviations: BMI, body mass index. [a] analyses were adjusted for age, sex, education, race/ethnicity, ratio of family income to poverty, smoking status, alcohol drinking and BMI except the stratification variable. MVPA, moderate-to-vigorous physical activity per week (minutes). *: $p < 0.05$.
(DOCX)

## Acknowledgments

The authors would like to acknowledge the support from all the team members and all staff of the National Center for Health Statistics.

**Disclaimer:** The findings and conclusions in this report are those of the authors and do not necessarily represent the official position of the Centers for Disease Control and Prevention.

## Author Contributions

**Conceptualization:** Qingsong Chen, Wenhan Yang.

**Data curation:** Xia Zeng, Jitian Huang.

**Formal analysis:** Qian Li.

**Methodology:** Qingsong Chen, Wenhan Yang.

**Project administration:** Tianran Shen.

**Software:** Yingxia Xu, Xiaofang Yan.

**Supervision:** Yingxia Xu, Yanmei Li.

**Validation:** Tianran Shen, Qian Li.

**Writing – original draft:** Xia Zeng, Jitian Huang.

**Writing – review & editing:** Xiaohui Xing.

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
