## [Decision Letter · Decision Letter 0]

30 Jan 2024

PONE-D-23-34240Nonlinear dose-response association of moderate-to-vigorous physical activity with hyperuricemia in US adults: NHANES 2007-2018PLOS ONE

Dear Dr. Yang,

Thank you for submitting your manuscript to PLOS ONE. After careful consideration, we feel that it has merit but does not fully meet PLOS ONE’s publication criteria as it currently stands. Therefore, we invite you to submit a revised version of the manuscript that addresses the points raised during the review process.

We look forward to receiving your revised manuscript.

Kind regards,

Yuichiro Nishida

Academic Editor

PLOS ONE

Journal Requirements:

2.Note from Emily Chenette, Editor in Chief of PLOS ONE, and Iain Hrynaszkiewicz, Director of Open Research Solutions at PLOS: Did you know that depositing data in a repository is associated with up to a 25% citation advantage (https://doi.org/10.1371/journal.pone.0230416)? If you’ve not already done so, consider depositing your raw data in a repository to ensure your work is read, appreciated and cited by the largest possible audience. You’ll also earn an Accessible Data icon on your published paper if you deposit your data in any participating repository (https://plos.org/open-science/open-data/#accessible-data).

Reviewers' comments:

Reviewer's Responses to Questions

**Comments to the Author**

1. Is the manuscript technically sound, and do the data support the conclusions?

Reviewer #1: No

Reviewer #2: Yes

2. Has the statistical analysis been performed appropriately and rigorously? 

Reviewer #1: No

Reviewer #2: Yes

3. Have the authors made all data underlying the findings in their manuscript fully available?

Reviewer #1: Yes

Reviewer #2: Yes

4. Is the manuscript presented in an intelligible fashion and written in standard English?

Reviewer #1: No

Reviewer #2: No

5. Review Comments to the Author

Reviewer #1: Zeng et al. aimed to explore the dose-response association of physical activity levels with hyperuricemia (HUA) by utilizing the National Health and Nutrition Examination Survey (NHANES, 2007–2018) data. They found that the low moderate-to-vigorous physical activity (MVPA) was associated with a higher odds ratio of HUA. Although the results seemed to be interesting, the reviewer has several concerns to be addressed.

1. Although hypertension was listed in the baseline characteristics, the authors did not adjust it. Aside from hypertension, many other comorbidities like diabetes, chronic kidney disease, heart failure, etc., are known to be highly correlated with HUA. These important covariates are not adjusted in the study.

2. To better delineate the relationship between MVPA and HUA, a time-to-event analysis is recommended.

3. Relevant biochemical data like fasting blood glucose, lipid profiles, glomerular filtration rate, etc., should be also included in the statistical models.

Reviewer #2: The results of this manuscript indicate that this study based on a nationally representative population of American adults found that compared with the moderate-to-vigorous physical activity (MVPA) level, the low MVPA level was associated with the higher risk of hyperuricemia (HUA). There was a-U-shaped dose-response relationship between MVPA level and HUA. The manuscript is interesting and sample size is sound. However, there are several concerns in it.

Major comments:

1) Page 9, lines 156-158: How did authors deal with sports time activity?

2) Page 9, lines 167-169: It might be a good idea to divide into 5 groups rather than 3 groups. Because the division of 5 groups may be a good shape when authors want to express a U-shaped relationship.

3) Page 17, Table 2: In model 3, there was not a significant positive association of HUA for participants in the high PA group: ≥3000 MET-min/wk compared to moderate group. Can authors say this is a U-shaped relationship among 3 groups?

4) Totally, some sentences are redundant. For examples; page 20, line 360-361: “To our knowledge, this was the first time” and page 21, line 382: “To the best of our knowledge, this was the first study” are almost same.

Minor comments:

1) Page 6, line 94: Rather than “Shu Yuan”, “Yuan S, et al.” may be better.

2) Page 6, line 98: “[15-17]” is better.

3) Page 7, line 113: “Methods” should be moved down one line.

4) Page 11, line 208: “groups.” is correct.

5) Page 11, line 219: “sex, and race/ethnicity” is correct.

6) Page 15, line 272: “BMI was” may be better.

7) Page 20, line 366: “Xiong Z, et al.” is correct.

8) Page 20, line 369: “elevation [41].” is correct.

9) Page 20, line 377: “previous studies” needs some references.

10) Page numbers of most of the references should be correctly omitted. For examples, Ref. “2177-80” may be correct.

11) Ref. 39: not “J Diabetes” but “J Diabetes.”

12) Page 30, line 585: not *: p≤0.05 but p<0.05.

6. PLOS authors have the option to publish the peer review history of their article (what does this mean?). If published, this will include your full peer review and any attached files.

Reviewer #1: No

Reviewer #2: No

---

## [Author Response · Author response to Decision Letter 0]

20 Mar 2024

Dear reviewers,

Thank you very much for your comments and professional advice concerning our manuscript entitled “Nonlinear dose-response association of moderate-to-vigorous physical activity with hyperuricemia in US adults: NHANES 2007-2018”(ID: PONE-D-23-34240). These opinions help to improve academic rigor of our article. Based on your suggestion and request, we have studied comments carefully and have made correction which we hope meet with approval. We hope that our work can be improved again. The main corrections in the paper and the responds to the reviewer’s comments are as flowing: 

Reviewer Comments:

Reviewer #1

Comments to the Author

Zeng et al. aimed to explore the dose-response association of physical activity levels with hyperuricemia (HUA) by utilizing the National Health and Nutrition Examination Survey (NHANES, 2007–2018) data. They found that the low moderate-to-vigorous physical activity (MVPA) was associated with a higher odds ratio of HUA. Although the results seemed to be interesting, the reviewer has several concerns to be addressed.

Response to comment: We would like to thank you for your detailed comments and valuable suggestions to improve the quality of our manuscript. We have revised specifically each suggestion below.

1.Although hypertension was listed in the baseline characteristics, the authors did not adjust it. Aside from hypertension, many other comorbidities like diabetes, chronic kidney disease, heart failure, etc., are known to be highly correlated with HUA. These important covariates are not adjusted in the study.

Response to comment: Special thanks to you for your good comments. In response to the critical covariates you highlighted, we have diligently incorporated them to the fullest extent possible and conducted a comprehensive sensitivity analysis. The outcomes of this analysis revealed that the prevalence of hypertension exhibited statistically significant differences across various groups. Conversely, the prevalence rates of diabetes, chronic kidney disease, and heart failure did not demonstrate statistically significant disparities among the groups (Table 1). Consequently, we have refined our model to integrate hypertension as a significant variable, and the manuscript has been updated accordingly with the pertinent data revisions. Regarding the additional important covariates you identified, upon careful consideration, we believe that their impact on our results appears to be relatively minor. Therefore, we opted not to adjust our model to include these covariates. We appreciate your insightful suggestions and have strived to address them with the utmost rigor and transparency in our revised analysis.

2.To better delineate the relationship between MVPA and HUA, a time-to-event analysis is recommended.

Response to comment: Thank you for your valuable suggestions. In our study, we employed a cross-sectional survey design, utilizing a large, nationally representative dataset from NHANES without subsequent follow-up investigations. Given the nature of a cross-sectional survey, we assert that the results derived without performing time-to-event analysis are both reasonable and conservative. We appreciate your suggestion regarding the implementation of time-to-event analysis. We will give this approach considerable priority in our future research endeavors, recognizing its potential to enhance the depth and quality of our findings.

3.Relevant biochemical data like fasting blood glucose, lipid profiles, glomerular filtration rate, etc., should be also included in the statistical models.

Response to comment: Thank you for your suggestion. In addressing the pivotal covariates you identified, we have endeavored to incorporate them to the greatest extent feasible. Regrettably, due to the absence of data on glomerular filtration rate within the NHANES database, we were unable to conduct further analyses on this parameter. Similarly, the NHANES database does not provide data on fasting blood glucose levels, compelling us to utilize serum glucose levels as a proxy for our subsequent sensitivity analysis. The results of this analysis indicated that the variations in serum glucose, triglycerides, and cholesterol across different groups did not reach statistical significance (Table 2). Consequently, we infer that these indicators likely exert a minimal impact on our findings, leading us to exclude them from adjustments in our model. We sincerely appreciate your insightful feedback and have rigorously attempted to address it within the constraints of the available data. Our commitment to transparency and scientific rigor has guided our approach in revising the analysis and manuscript accordingly.

Reviewer Comments:

Reviewer #2

Comments to the Author

The results of this manuscript indicate that this study based on a nationally representative population of American adults found that compared with the moderate-to-vigorous physical activity (MVPA) level, the low MVPA level was associated with the higher risk of hyperuricemia (HUA). There was a-U-shaped dose-response relationship between MVPA level and HUA. The manuscript is interesting and sample size is sound. However, there are several concerns in it.

Response to comment: We would like to thank you for providing us with further suggestions and guidance. We have revised our manuscript following your suggestions, as illustrated below. We hope that our efforts have succeeded in allaying you concerns and would be grateful for any further guidance. 

Major comments:

1.Page 9, lines 156-158: How did authors deal with sports time activity?

Response to comment: Thank you for your question. According to the Global Physical Activity Questionnaire, we classify sports time activity as vigorous leisure activity or moderate leisure activity based on their intensity, which suggested MET scores in NHANES is 4 or 8.

2.Page 9, lines 167-169: It might be a good idea to divide into 5 groups rather than 3 groups. Because the division of 5 groups may be a good shape when authors want to express a U-shaped relationship.

Response to comment: Thanks for your valuable counsel. Since the survey of physical activity level in the database adopts the global physical activity questionnaire, we used the classification standard in the article "Validity and Reliability of the Global Physical Activity Questionnaire (GPAQ)" for reference to the number of groups of physical activity level (Total activity in MET-min/week), and divided them into 3 groups instead of 5 groups. Besides, we used RCS curves to fit the U-shaped relationship between them. MVPA levels is a quantitative data based on individual rather than grouping variable data in RCS model. Therefore, we believe that the number of groups may have little effect on the expression of a U-shaped relationships.

3.Page 17, Table 2: In model 3, there was not a significant positive association of HUA for participants in the high PA group: ≥3000 MET-min/wk compared to moderate group. Can authors say this is a U-shaped relationship among 3 groups?

Response to comment: Many thanks for your comment. We describe the U-shaped relationship between them mainly based on the RCS curve drawn based on large sample quantitative data. The RCS curve showed a U-shaped trend. As stated by the reviewer, there was not a significant positive association of HUA for participants in the high PA group compared to moderate group. Consequently, taking into account the insights provided by the reviewer, we have refined the articulation regarding the U-shaped curve's depiction within our manuscript to ensure a more nuanced and professional presentation.

4.Totally, some sentences are redundant. For examples; page 20, line 360-361: “To our knowledge, this was the first time” and page 21, line 382: “To the best of our knowledge, this was the first study” are almost same.

Response to comment: Thank you for your significant reminding. We have made corrections and removed the redundant sentences.

Minor comments:

1.Page 6, line 94: Rather than “Shu Yuan”, “Yuan S, et al.” may be better.

Response to comment: Thank you for your significant reminding. We have rewritten “Yuan S, et al.” instead of “Shu Yuan”.

2.Page 6, line 98: “[15-17]” is better.

Response to comment: Thank you for your significant reminding. We have rewritten “[15-17]” instead of “[15,16,17]”.

3.Page 7, line 113: “Methods” should be moved down one line.

Response to comment: Thank you for your significant reminding. We have made an adjustment.

4.Page 11, line 208: “groups.” is correct.

Response to comment: Thank you for your significant reminding. We have rewritten “groups.” instead of “groups .”.

5.Page 11, line 219: “sex, and race/ethnicity” is correct.

Response to comment: Thank you for your significant reminding. We have rewritten “sex, and race/ethnicity” instead of “sex, race/ethnicity”.

6.Page 15, line 272: “BMI was” may be better.

Response to comment: Thanks for your valuable counsel. We have rewritten “BMI was” instead of “BMI were”.

7.Page 20, line 366: “Xiong Z, et al.” is correct.

Response to comment: Thank you for your significant reminding. We have rewritten “Xiong Z, et al.” instead of “Xiong Z et al”.

8.Page 20, line 369: “elevation [41].” is correct.

Response to comment: Thanks for your valuable counsel. We have rewritten “elevation [41].” instead of “elevation [41]”.

9.Page 20, line 377: “previous studies” needs some references.

Response to comment: Thanks for your valuable counsel. We have attached some references.

10.Page numbers of most of the references should be correctly omitted. For examples, Ref. “2177-80” may be correct.

Response to comment: Thank you for your significant reminding. We have made some changes to address this issue. We have rewritten “2177-80” instead of “2177-2180”. We have rewritten “991-9” instead of “991-999”. We have rewritten “291-9” instead of “291-299”. We have rewritten “272-82” instead of “272-282”. We have rewritten “221-35” instead of “221-235”. We have rewritten “147-54” instead of “147-154”. We have rewritten “602-04” instead of “602-604”.

11.Ref. 39: not “J Diabetes” but “J Diabetes.”

Response to comment: Thank you for your significant reminding. We have rewritten “J Diabetes.” instead of “J Diabetes”.

12.Page 30, line 585: not *: p≤0.05 but p<0.05.

Response to comment: Thanks for your valuable counsel. We have rewritten “p< 0.05” instead of “p≤ 0.05”.

We tried our best to improve the manuscript and some changes in the manuscript. These changes will not influence the content and framework of the paper. We appreciate for reviewer’s warm work earnestly, and hope that the correction will meet with approval.

Once again, thank you very much for your comments and suggestions.

Yours sincerely,

---

## [Decision Letter · Decision Letter 1]

3 Apr 2024

Nonlinear dose-response association of moderate-to-vigorous physical activity with hyperuricemia in US adults: NHANES 2007-2018

PONE-D-23-34240R1

Dear Dr. Yang,

We’re pleased to inform you that your manuscript has been judged scientifically suitable for publication and will be formally accepted for publication once it meets all outstanding technical requirements.

Kind regards,

Yuichiro Nishida

Academic Editor

PLOS ONE

Additional Editor Comments (optional):

Reviewers' comments:

Reviewer's Responses to Questions

**Comments to the Author**

1. If the authors have adequately addressed your comments raised in a previous round of review and you feel that this manuscript is now acceptable for publication, you may indicate that here to bypass the “Comments to the Author” section, enter your conflict of interest statement in the “Confidential to Editor” section, and submit your "Accept" recommendation.

Reviewer #2: All comments have been addressed

2. Is the manuscript technically sound, and do the data support the conclusions?

Reviewer #2: Yes

3. Has the statistical analysis been performed appropriately and rigorously? 

Reviewer #2: Yes

4. Have the authors made all data underlying the findings in their manuscript fully available?

Reviewer #2: Yes

5. Is the manuscript presented in an intelligible fashion and written in standard English?

Reviewer #2: Yes

6. Review Comments to the Author

Reviewer #2: The revised manuscript has been much improved according to the reviewers' helpful comments.

This reviewer does not have any further comments.

7. PLOS authors have the option to publish the peer review history of their article (what does this mean?). If published, this will include your full peer review and any attached files.

Reviewer #2: No

---

## [Editor Report · Acceptance letter]

14 May 2024

PONE-D-23-34240R1 

PLOS ONE

Dear Dr. Yang, 

I'm pleased to inform you that your manuscript has been deemed suitable for publication in PLOS ONE. Congratulations! Your manuscript is now being handed over to our production team.

Kind regards, 

on behalf of

Dr. Yuichiro Nishida 

Academic Editor

PLOS ONE